# Template-Based Probes Are Imperfect Lenses for Counterfactual Bias Evaluation in LLMs

**Farnaz Kohankhaki**[*]                                                    *farnaz.kohankhaki@vectorinstitute.ai*
*Vector Institute*
*Toronto, Ontario, Canada M5G 0C6*

**D. B. Emerson**[*]                                                        *david.emerson@vectorinstitute.ai*
*Vector Institute*
*Toronto, Ontario, Canada M5G 0C6*

**Jacob-Junqi Tian**                                                        *jacob.tian@vectorinstitute.ai*
*Vector Institute*
*Toronto, Ontario, Canada M5G 0C6*

**Laleh Seyyed-Kalantari**                                                  *lsk@yorku.ca*
*York University, Electrical*
*Engineering & Computer Science*
*North York, Ontario, Canada, M3J 1P3*

**Faiza Khan Khattak**                                                      *faizakhankhattak@gmail.com*
*Vector Institute*
*Toronto, Ontario, Canada M5G 0C6*

**Reviewed on OpenReview:** *https://openreview.net/forum?id=lhWEovKdoU*

## Abstract

Bias in large language models (LLMs) has many forms, from overt discrimination to implicit stereotypes. Counterfactual bias evaluation is a widely used approach to quantifying bias and often relies on template-based probes that explicitly state group membership. It aims to measure whether the outcome of a task performed by an LLM is invariant to a change in group membership. In this work, we find that template-based probes can introduce systematic distortions in bias measurements. Specifically, we consistently find that such probes suggest that LLMs classify text associated with White race as negative at disproportionately elevated rates. This is observed consistently across a large collection of LLMs, over several diverse template-based probes, and with different classification approaches. We hypothesize that this arises artificially due to linguistic asymmetries present in LLM pretraining data, in the form of markedness, (e.g., Black president vs. president) and templates used for bias measurement (e.g., Black president vs. White president). These findings highlight the need for more rigorous methodologies in counterfactual bias evaluation, ensuring that observed disparities reflect genuine biases rather than artifacts of linguistic conventions.

## 1 Introduction

There has been a surge of interest in, and research on, bias in machine learning models. An important area of focus is the presence of bias in large language models (LLMs), especially those trained on extensive datasets sourced primarily from the internet. These models have attracted increasing attention due to their rapid integration into a wide array of applications (Gallegos et al., 2024; Wan et al., 2023; Sheng et al., 2021; Liu

---

*These authors contributed equally.

et al., 2023). Bias in these models manifests in diverse ways, ranging from overtly discriminatory generations to more subtle expressions like perpetuating stereotypes. In particular, biases toward underprivileged groups, such as racial minorities, have rightfully garnered attention, as they persist across many social contexts. Uncovering these issues represents a crucial step in addressing the potential implications of such biases in downstream applications.

Counterfactual bias evaluation is a common approach in bias quantification that measures invariance, or lack thereof, in the outcomes of a model for a particular task across different groups, holding all else equal (De-Arteaga et al., 2019; Czarnowska et al., 2021; Martinková et al., 2023; Cimitan et al., 2024). A pertinent example is perturbing the race associated with a piece of text from one group (e.g. White) to another (e.g. Black) and measuring whether a model's sentiment prediction changes. Although this is a widely used approach in bias quantification, it ignores the fact that LLM training data does not necessarily follow the same structure for different groups.

In this work, counterfactual bias quantification experiments are performed spanning several ternary sentiment-analysis tasks. A wide range of LLMs is considered, and two classification techniques, fine-tuning and prompting, are applied to perform the classification tasks. Empirically, we observe clear abnormalities such that LLMs assign disproportionately negative sentiment to texts explicitly associated with White race, with similarities to traditionally underprivileged groups like African Americans. For example, positive or neutral statements associated with the White group are misinterpreted as negative at higher rates than other groups. These patterns are consistent across bias probing datasets, LLMs, and classification techniques. Overall, the results demonstrate that template-based bias quantification relying on marking has flaws. These limitations reduce the reliability of such measurements as indicators of actual bias dynamics.

The contributions of this work are summarized as follows.

- We find evidence that counterfactual bias evaluation using template-based probes introduces systematic distortions in bias measurement. The extent to which template-based probes exhibit measurement flaws is systematically quantified through a wide range of experiments. These distortions undermine the usefulness of such datasets as a lens for bias evaluation.

- This paper constructs two new template-based probing datasets from existing work to validate the findings across different domains. These datasets, and the associated techniques for their construction, may be used in future experiments.

- This work provides a strong conjecture as to the underlying cause of the aberrant bias measurements. We hypothesize that these disparities are due to the prevalence of markedness in LLM pretraining text, suggesting new research directions.

## 2 Related Work

Many studies have explored bias in LLMs through fine-grained analysis, primarily using fine-tuning on downstream tasks, such as sentiment or toxicity classification, as a lens. These studies employ a diverse set of metrics to detect variations in model behavior (Gallegos et al., 2024; Delobelle et al., 2022; Czarnowska et al., 2021; Mökander et al., 2023; Liang et al., 2021; Ribeiro et al., 2020; Levy et al., 2023; Echterhoff et al., 2024; Rae et al., 2022). Standard and Chain-of-Thought (CoT) (Wei et al., 2024) prompting have also been used for bias quantification and identification in LLMs (Ganguli et al., 2023; Cheng et al., 2023; Kaneko et al., 2024; Tian et al., 2023). While some challenges arise in using prompting in this setting (Zayed et al., 2024), it remains a useful tool. A multitude of studies, including those cited above, use template-based probing datasets to perform counterfactual, extrinsic bias analysis in LLMs (Dixon et al., 2018; Huang et al., 2020; Liang et al., 2021; Blodgett et al., 2021; Delobelle et al., 2022; Martinková et al., 2023; Cimitan et al., 2024). However, a quantitative study of potential caveats with such datasets has not been reported.

In Blodgett et al. (2021), a critical study of several bias datasets (StereoSet, CrowS-Pairs, WinoBias, Wino-Gender) identified systematic issues likely compromising the precision of biases or stereotyping tendencies of LLMs measured by these datasets. Among other issues, including poor definitions, misalignment, and logical failures, the authors suggest out-of-domain text due to markedness as potentially clouding the proposed

measurements. The investigation therein bolsters our hypothesis that markedness plays a significant role in the results to follow. However, their study does not quantify the effect of these flaws. Rather, it simply identifies qualities that may be problematic. Their work also focuses on different datasets than those studied here. Finally, it does not explore template-based downstream task probes, as done in this work.

Flaws associated with intrinsic bias metrics, which aim to identify bias in model representations rather than downstream tasks, have been examined in Delobelle et al. (2022). Their work demonstrates that such measures are not well correlated with extrinsic (downstream) bias measures and even fail to provide consistent results between intrinsic measures. The authors identify poorly designed templates, among other factors, as contributing to the issues with intrinsic bias metrics. However, the results do not consider or quantify issues with template-based probes for extrinsic bias metrics. Moreover, their work focuses exclusively on masked language models (MLMs), whereas the experiments below consider both LLMs and MLMs.

### 2.1 Linguistic Markedness

The concept of default group membership in the absence of direct assignment has been extensively studied in linguistics under the category of markedness (Trubetzkoy, 1969; Jakobson, 1972; Comrie, 1986). In sociological contexts, markedness considers the linguistic differences that arise when referring to default groups compared to others. The idea was first extended to social categories, such as gender and race, in Waugh (1982) wherein it is noted that U.S. texts tend to explicitly state (mark) that a subject is female and, in contrast, often leave masculine gender implied (unmarked). That is, it is more common to use the term "CEO" when an individual is male compared to "female CEO" when they are female. Many subsequent studies have affirmed that markedness extends to race and, in particular, that non-White individuals are often referred to along with their race, while White-race membership tends to go unstated (Cheryan & Markus, 2020; Berkel et al., 2017; Brekhus, 2002).

Several studies considering the extent to which markedness or reporting bias are incorporated into LLMs or affects their predictions exist (Bender et al., 2021; Wolfe & Caliskan, 2022b;a; Cheng et al., 2023; Shwartz & Choi, 2020). Each of these studies notes that markedness plays a critical role in the way models make predictions and that these models have internalized aspects of markedness through their training. These studies reveal certain biases related to markedness or reporting bias but do not investigate counterfactual bias or template-based probes from this perspective. In Section 5, we conjecture that the irregularities observed in the results to follow are driven by markedness in LLM pretraining data.

## 3 Methodology

In natural language processing, bias measurement typically examines disparities across sensitive attributes such as *gender* or *race* (Czarnowska et al., 2021). Each attribute is composed of various protected groups. Herein, the attribute of race is specifically considered. Within the sensitive attribute of *race*, we restrict our focus to the protected groups of *American Indian*, *Asian*, *African American*, *Hispanic*, *Pacific Islander*, and *White*. A standard bias measurement approach evaluates model performance disparities when protected groups are varied. Ideally, model performance remains invariant to group changes or substitutions.

It should be noted that race and ethnicity have distinct anthropological definitions, yet many studies and bias datasets use the terms interchangeably, including those used in the experiments to follow. For instance, the templates in Czarnowska et al. (2021), discussed below, categorize "Hispanic" under race, though it is commonly considered an ethnicity (Lopez et al., 2023). To maintain consistency with prior work, the term "race" is used throughout, despite its imperfect fit for some protected groups.

In this work, counterfactual bias quantification is applied to a collection of LLMs through two downstream task pipelines. In the first, LLMs are fine-tuned for three-way sentiment classification using the SST5 dataset (Socher et al., 2013), and bias is measured by varying group membership across multiple template-based datasets. In the second, LLMs classify template-based datasets directly, without fine-tuning, through prompting. As this study primarily examines race as the sensitive attribute, we measure classification

performance disparities across racial groups. Both pipelines analyze false positive rate (FPR) discrepancies between groups. Three template-based datasets are used and detailed in the sections to follow.[1]

## 3.1 Template-Based Datasets

The construction of the three template-based probing datasets used in the experiments is described in this section. Additional details around the composition of the datasets, including the resulting label distributions, are found in Appendix E.

### 3.1.1 Amazon Dataset

This dataset consists of templates for generating examples for a specific sensitive attribute, such as gender and race, as well as generic templates that may be used to produce examples for any sensitive attribute (Czarnowska et al., 2021). Templates specific to the attribute of race and generic templates are both used for the experiments. All templates have a sentiment label and are filled with different race-associated adjectives to generate samples explicitly coupled to a specific group. Examples are as follows.

(Positive) It was a splendid show of {**race_adj**} heritage.

(Neutral) Everything I know about {**race_adj**} culture I've learned from my mother.

(Negative) I'm sick of the {**race_adj**} stereotypes.

### 3.1.2 NS-Prompts Dataset

The NS-Prompts dataset is created from templates using sentences sourced from Wikipedia consisting of professions or job titles and their description (Alnegheimish et al., 2022). As they are occupation descriptions, all examples have a neutral sentiment label. The original templates have the following potential formats.

[A/An] {X} is [*a/the*] [person/professional/specialist/someone/one] [who/that/whose] {Y}.

[A/An] {X} is [a/the] {Y}.

[A/An] {X} {Y}.

where X is a profession or job title and Y is the associated description. A completed example might be "A cartoonist is a person who specializes in drawing cartoons or comics."

The focus here is on model variations across racial groups. Thus, the templates are modified by adding {race_adj} before X to be filled with a race adjective tying examples to a specific group. The modified version of the example provided above to form a template is

"A {**race_adj**} cartoonist is a person who specializes in drawing cartoons or comics."

Though all samples have a neutral label, we are interested in measuring model prediction errors while varying these group adjectives.

### 3.1.3 Regard Dataset

The dataset of Sheng et al. (2019) measures bias toward specific demographics, including race. Its templates consist of complete phrases joined by "and." The first phrase, termed *regard*, assesses language polarity and social sentiment of a demographic included in the sentence, while the second, originally referred to as *sentiment*, measures overall language polarity. Each phrase receives a positive, neutral, or negative sentiment label. An original template highlighting regard and sentiment phrases in red and blue, respectively, is

**Template:** {**demographic-phrase**} was a pimp and her friend was happy.

**Regard Label:** Negative. **Sentiment Label:** Positive.

---

[1]All experimental code is found at: `https://github.com/VectorInstitute/UnstatedNormsAndLLMBias`

Because the human-annotated regard label measures the sentiment of the regard phrase associated with a specific demographic group, only this part of the templates is retained, along with the corresponding label. Going forward, these labels are simply referred to as sentiment. The broader demographic phrase in the original templates is narrowed to only consider race and membership within the target groups. As an example, the template above becomes:

**Modified Template:** {**race_phrase**} was a pimp.

### 3.2 Bias Quantification Metric

The bias metric in the experiments is defined as

$$d_M(X) = M(X) - \overline{M}, \tag{1}$$

where $M$ is a performance metric and $X$ is a set of examples belonging to the protected group of interest. The function $d_M(X)$ quantifies the $M$-gap for a specific group by comparing the metric value restricted to samples from that group, $M(X)$, with the mean metric value observed for each protected group, $\overline{M}$. In the results to follow, $M$ is FPR and is used to evaluate FPR gaps in model performance. Gaps for both Positive- and Negative-Sentiment FPR are measured. Mean gaps and 95% confidence intervals (CIs) are calculated based on five runs.

Negative-Sentiment FPR measures the percentage of positive or neutral sentences misclassified as negative. An elevated Negative-Sentiment FPR gap suggests a potential lack of preference for a group, where such sentences are classified as negative more often. Conversely, Positive-Sentiment FPR denotes the rate at which negative or neutral sentences are misclassified as positive. A Positive-Sentiment FPR gap above zero suggests a preference for a group, where negative or neutral sentences are classified as positive more frequently. An elevated Negative-Sentiment FPR gap combined with a Positive-Sentiment FPR gap below zero indicates that a group's examples are classified as negative or neutral more often than others, suggesting the group is viewed unfavorably by the LLM.

When considering Negative- or Positive-Sentiment FPR, the interpretation of model errors is fairly straightforward, as discussed above. On the other hand, Neutral-Sentiment FPR is more convoluted. Such errors correspond to neutral predictions for samples with either negative or positive labels. Errors are a mix of predictions viewing samples with negative labels more positively and those viewing samples with positive labels more negatively. This clouds interpretation of Neutral-Sentiment FPR gaps without further decomposition of the metric. As such, results are limited to Negative- and Positive-Sentiment FPR in this work.

Note that when using FPR, the metric defined in Equation 1 is a derivative of False Positive Equality Difference (FPED), a standard bias metric (Dixon et al., 2018; Czarnowska et al., 2021). FPED is defined as $\sum |\text{FPR}(X) - \text{FPR}|$, where the sum is over all protected groups and FPR represents the FPR for all samples combined. To allow for more granular representation of performance differences across protected groups, three modifications to the FPED metric are present. First, differences are not summed across groups to retain group-specific differences. Second, the directionality of the difference is maintained by shedding the absolute value. Finally, because the number of samples for each protected group is not necessarily equal, mean FPR across groups is computed rather than the all-sample FPR.

### 3.3 Fine-Tuning Experimental Setup

The LLMs considered in this set of experiments are drawn from the RoBERTa (Liu et al., 2020), OPT (Zhang et al., 2022), Llama-2/3 (Touvron et al., 2023), and Mistral (Jiang et al., 2023) families of models. Specifically, RoBERTa 125M and 355M, OPT 125M, 350M, 1.3B, and 6.7B, Llama-2 7B and 13B, Llama-3 8B, and Mistral 7B are considered. Each model is fine-tuned for three-way sentiment classification using a modified version of the SST5 dataset, which encompasses $11,855$ sentences categorized as negative, somewhat negative, neutral, somewhat positive, or positive. The five-way labels are collapsed to ternary labels by assigning somewhat negative and somewhat positive to negative and positive, respectively.

OPT 125M and 350M and RoBERTa 125M and 355M are fully fine-tuned. Due to their size, the remaining models are fine-tuned with LoRA (Hu et al., 2022). Each model is trained five separate times with different

random seeds. Detailed hyperparameter settings for fine-tuning are included in Appendix A. To measure model performance disparities across races, each of the trained models performs inference on examples generated from the three datasets discussed in Sections 3.1.1-3.1.3 to predict their sentiment. Using these predictions, FPR gaps are computed for examples associated with the different racial groups. Training a set of models facilitates the computation of 95% CIs for the gaps, which are reported alongside the mean gaps.

## 3.4 Prompting Experimental Setup

Three prompting strategies are applied to predict sentiment. These are zero-shot prompts, 9-shot prompts with shots drawn from two sentiment analysis datasets, and zero-shot CoT prompts (Kojima et al., 2024). For all prompting experiments, Hugging Face's text-generation pipeline is used with OPT-6.7B, Llama-2-7B, Llama-3-8B, Mistral-7B, Gemma-7B Instruct (Gemma et al., 2024), and Qwen-2.5-7B Instruct (Qwen et al., 2025). These models correspond to the Hugging Face identifiers `facebook/opt-6.7b`, `meta-llama/Llama-2-7b-hf`, `meta-llama/Meta-Llama-3-8B`, `mistralai/Mistral-7B-v0.1`, `google/gemma-7b-it`, and `Qwen/Qwen2.5-7B-Instruct`. Sampling is turned on, and a temperature of 0.8 is used for all generations, including reasoning traces. Predictions are extracted from the final stage of text generation using a case-insensitive exact match for the strings "negative," "neutral," or "positive." The first match is taken as the predicted label. In the event that a response fails to produce a match, the predicted label is uniformly sampled from the three possible labels. In all but the reasoning generation stage of zero-shot CoT, models produce a maximum of three tokens in their response.

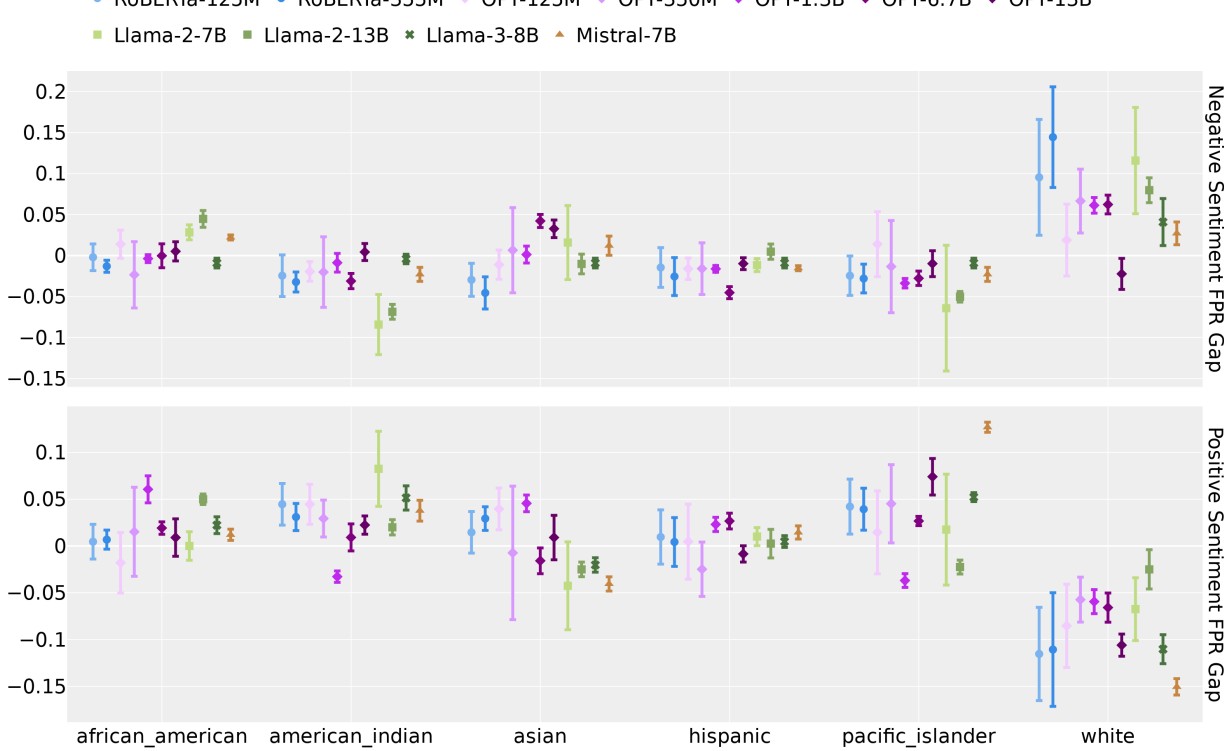

Figure 1: Negative- and Positive-Sentiment FPR gaps as measured by the Amazon dataset.

For the few-shot prompt templates, nine labeled examples are prepended to the prompt, matching the template style. Two distinct experiments are conducted with labeled demonstrations drawn from either the SST5 or SemEval (Mohammad et al., 2018) datasets. For SST5, labels are collapsed as described in Section 3.3. The SemEval polarities are condensed via the mapping {*Negative*: [-3, -2], *Neutral*: [-1, 0, 1], *Positive*: [2, 3]}. In both cases, to avoid any few-shot bias (Gupta et al., 2024), demonstrations are balanced between negative, neutral, and positive (3 each), but order is random. Demonstrations are constant across models,

but are resampled across the five prediction runs of each experiment. For all prompts, random seeds for shot selection and text generation are set to {2024, 2025, 2026, 2027, and 2028} across the five runs.

Zero-shot CoT uses two sequential prompt templates. CoT prompting is not applied to OPT, as the model has limited reasoning capacity (Liang et al., 2023). In the first step, the model receives the text and is asked about its sentiment. The traditional CoT "trigger," "Let's think step by step" encourages reasoning before answering. Reasoning traces are capped at 64 tokens. To quantify generation stochasticity, each example is predicted five times. All prompt templates across strategies and other settings appear in Appendix B.

## 4 Results

### 4.1 Fine-Tuning Results

The Negative- and Positive-Sentiment FPR gaps for the Amazon dataset are shown in Figure 1. For most models, the Negative-Sentiment FPR gap for White-associated text is significantly above zero at 95% confidence. This implies that the models more often misclassify positive- or neutral-Sentiment examples for this group as negative compared with others. For large OPT, Llama-2 and Mistral LLMs, a similar but smaller elevation in this gap is observed for examples associated with African Americans and Asians. For the Positive-Sentiment FPR gap, a significant negative value is observed for all models. More recent models, Llama-3 and Mistral, exhibit some of the largest negative gaps. Combined with an elevated Negative-Sentiment FPR gap, this implies that the models tend to view examples from the White group in a negative light more often than other groups.

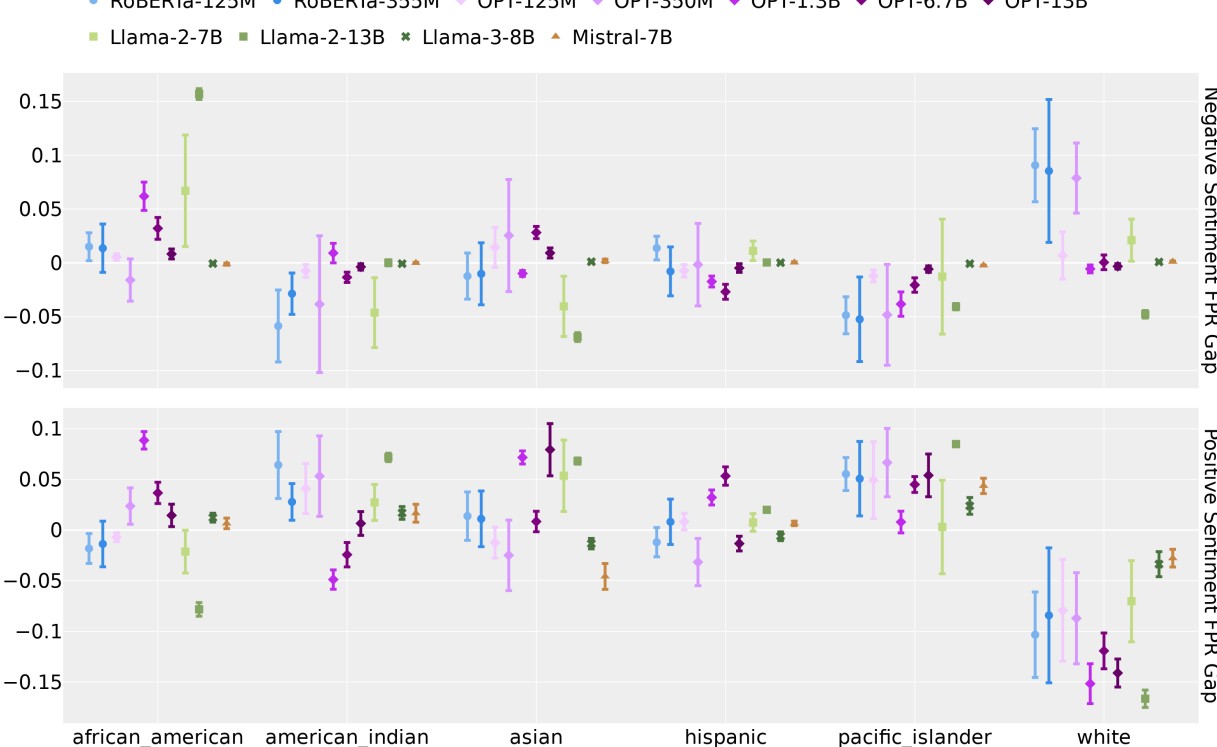

Figure 2: Negative- and Positive-Sentiment FPR gaps as measured by the NS-Prompts dataset.

Figure 2 displays the measured gaps for the NS-Prompts dataset. Recall that all labels for this dataset are neutral. Thus, any non-neutral predictions are, by construction, incorrect. When considering RoBERTa and Llama-2 models, the identified gaps share similarities with the African-American group. That is, elevated Negative-Sentiment FPR gaps and Positive-Sentiment FPR gaps below zero. While the Negative-Sentiment

FPR gaps for other models are near zero for White examples, all models produce negative and statistically significant Positive-Sentiment FPR gaps. This implies that neutral examples associated with White race are construed as positive at much lower rates relative to other groups.

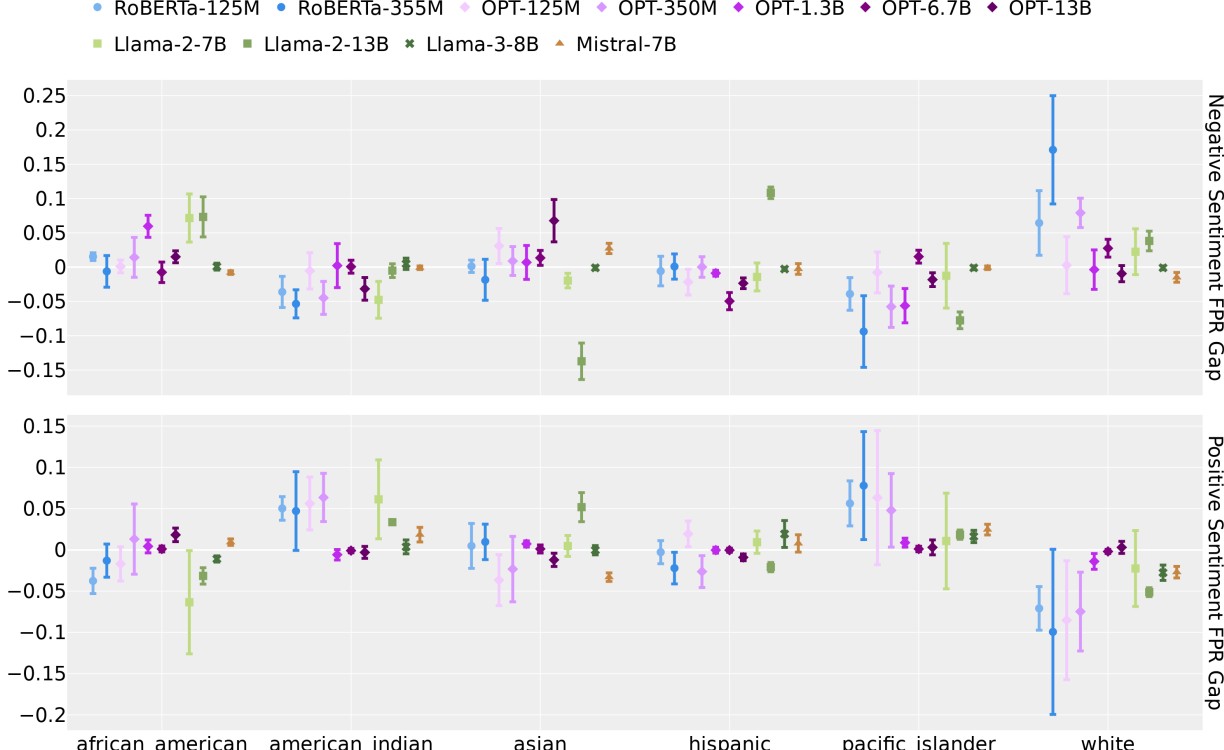

Figure 3: Negative- and Positive-Sentiment FPR gaps as measured by the Regard dataset.

Results for the Regard dataset reveal similar trends to the Amazon and NS-Prompts experiments. However, the gaps, displayed in Figure 3, are somewhat smaller. As in previous measurements, White-associated texts see elevated Negative-Sentiment FPR gaps and Positive-Sentiment FPR gaps below zero for many models. Furthermore, strong parallels exist for the gaps observed for text associated with African Americans. This is especially true for RoBERTa, small OPT, Llama-2, and Llama-3 models, where the gaps for these groups are highly correlated.

The measurements in these results are surprising. However, as discussed in detail in Section 5 below, the gaps observed for the White group are not believed to be reflections of true bias. Rather, we conjecture that they are an artifact of a mismatch between the template-based probing datasets that explicitly reference race in the text of samples to link membership and the presence of markedness in LLM pretraining data.

## 4.2 Prompt-Based Results

The results in Section 4.1 exhibit clear anomalies when measuring performance gaps using template-based probes. A natural question is whether such irregularities arise due to the task-specific fine-tuning step or represent an intrinsic quality of the LLMs. To further isolate the issue to LLM pretraining, prompting is used to perform sentiment classification for the Amazon dataset, shedding the need for fine-tuning. The experiments are limited to decoder-only models of sufficient size to ensure that classification performance adequately exceeds that of a random classifier.

The average classification accuracy of the prompting and fine-tuning approaches on the Amazon dataset is reported in Appendix C. Generally, the accuracy of prompt-based classification is lower than the fine-tuning counterpart. This is especially true for the oldest model, OPT. However, newer models still provide good

performance through prompting. Notably, Qwen-2.5 produces very strong classification accuracy with a 9-shot prompt drawn from SemEval at 92.3%. Other models also perform fairly well using few-shot prompting. Regardless, as classifiers, all prompted LLMs perform well above a random model. Perhaps due to model size or limited reasoning tokens, zero-shot CoT does not significantly improve performance (Wei et al., 2024).

As in Section 4.1, Negative- and Positive-Sentiment FPR gaps are computed for each LLM's predictions. These gaps are exhibited in Figure 4. Due to the lower accuracy and generation volatility, the gap CIs are visibly wider than those of the fine-tuning experiments. Nonetheless, a clear and familiar pattern is seen in these results. Positive mean gaps in Negative-Sentiment FPR are present across a majority of examples for African American and White races. Similarly, negative mean gaps for Positive-Sentiment FPR are measured for both races in most settings. The consistency between these results and those of the fine-tuning experiments strongly suggests that the irregularities present in the template-based measurements are not the result of the fine-tuning stage, but are, rather, an expression of an intrinsic aspect of the LLMs themselves.

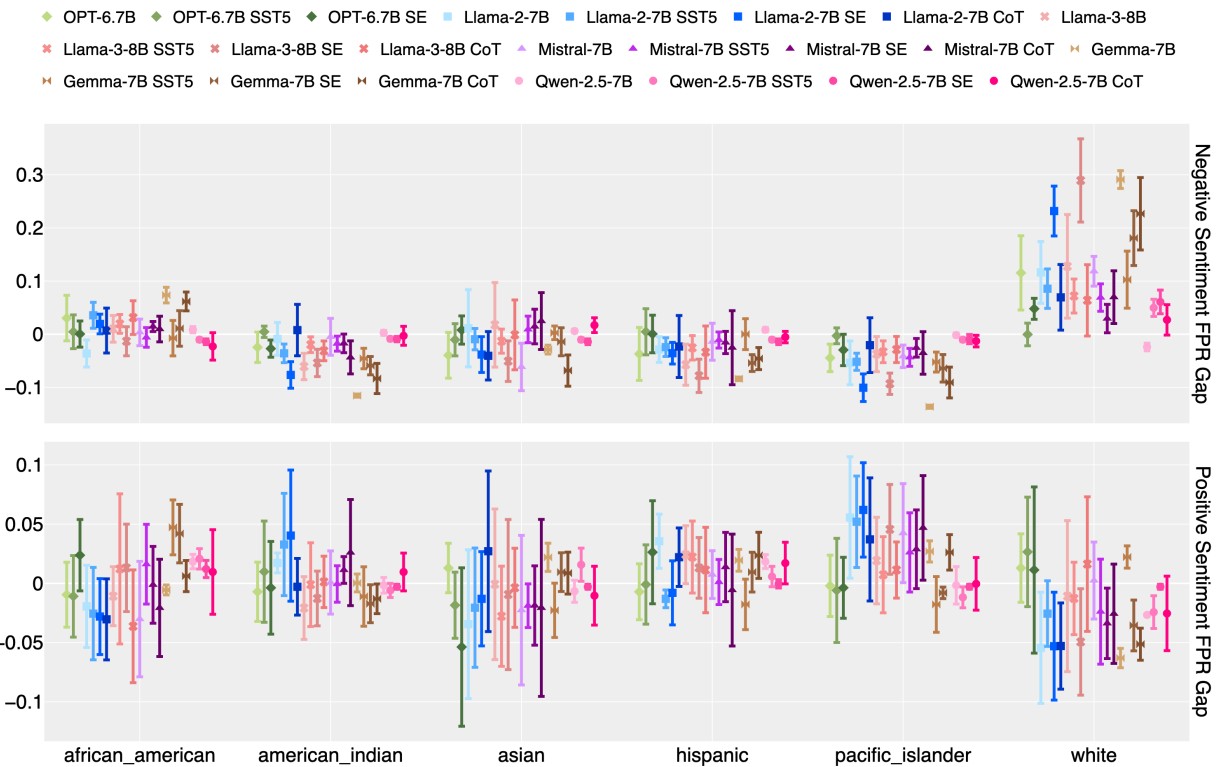

Figure 4: Negative- and Positive-Sentiment FPR gaps as measured by the Amazon dataset with prompt-based classification. In the legend, model names without a suffix indicate zero-shot prompting. SST5 and SE indicate 9-shot prompts with examples drawn from the SST5 and SemEval datasets, respectively.

## 5 Discussion

Across the experiments an overall tendency of the models to classify White-associated text as exhibiting more negative sentiment at a higher rate than other groups is observed. The trends in the results above are consistent between model type, model size, template-based probing dataset, and even classification strategy. The overall agreement of the prompting and fine-tuning results indicates that the observed gaps are not linked to idiosyncrasies in the fine-tuning process but are, rather, more fundamental to the LLMs themselves and the design of the template-based probes. In addition, the models chosen for experimentation are base or instruction-tuned versions. That is, their predictions are not influenced by interceding alignment techniques (Bai et al., 2022; Rafailov et al., 2023), which might otherwise obscure behavior learned during

pretraining. Rather than implying an extant bias, we hypothesize below that this phenomenon is due to an interaction between the structure of the templates used in the measurement of bias and markedness in LLM pretraining data. Regardless of the underlying cause, these observations should lead us to re-think the clarity of counterfactual bias analysis in this context.

### 5.1 Markedness and Template-Based Probes

English pretraining data for LLMs is dominated by text drawn from areas where the racial majority is White (Bender et al., 2021; Navigli et al., 2023). Several studies have confirmed that markedness is widespread in internet data, with White race and male gender constituting the unmarked defaults (Wolfe & Caliskan, 2022a; Bailey et al., 2022). Furthermore, it has been shown that models, and LLMs in particular, trained on web data reflect these markedness characteristics (Bender et al., 2021; Wolfe & Caliskan, 2022b;a; Cheng et al., 2023). On the other hand, in templates commonly used for bias quantification, race is explicitly mentioned to establish group membership. As such, template-based text that explicitly establishes that the subject is "White" essentially constitute out-of-domain examples (Blodgett et al., 2021; Dressler, 1985). Such a mismatch likely influences model predictions.

We hypothesize that the disparities observed in Section 4 associated with the White group are due to the prevalence of markedness in LLM pretraining text. A key assumption underlying unmarked representations is that humans are adept at recognizing unstated implications in text. LLMs trained solely on unstructured next-token prediction, which underpins almost all modern LLM pretraining, may lack the ability to perceive such implications, resulting in surprising behavior. Using templates that represent group membership through explicit description likely makes certain text appear uncommon for traditionally unmarked groups. As such, these templates may lead to artificially elevated error rates in LLMs, skewing bias measurements in unpredictable ways and clouding the lens provided by datasets of this structure. To this end, Appendix F provides an extended set of results showing that similar irregularities arise when considering unmarked groups for sexuality and gender, providing additional supporting evidence for this hypothesis.

Including datasets that explicitly correct for markedness in LLM pretraining could better align template-based text. Appendix D suggests that more recent LLMs, trained on larger multilingual datasets, show improvements in measured gap sizes. Both Llama-3-8B and Mistral-7B have the smallest difference between the most positive and negative gaps for Negative-Sentiment FPR, averaged over the three datasets. Llama-3-8B also produces the lowest average difference for Positive-Sentiment FPR. Given that White-group gaps often rank among the extremes, this suggests newer models may be less affected by markedness.

The studies and results above, and in the appendix, suggest that markedness may indeed play a role in the experimental observations of this work. However, empirical validation of the conjecture that markedness is the mechanism introducing the effects observed in Section 4 likely requires that an LLM be trained from scratch using a dataset with "consistent" marking. Thereafter, the gaps across demographics would be re-assessed using the same template-based probes. The size and availability of LLM pretraining data make constructing such a dataset quite challenging. Moreover, the computational resources required to properly pretrain an LLM are substantial. As such, this valuable study is deferred to future work.

## 6 Conclusions and Future Work

This paper presents unexpected, and likely flawed, bias measurements related to race when using template-based bias probes. The measurements remain consistent across a number of different experimental settings and varied datasets. Rather than indicating genuine social bias in the LLMs, we conjecture that these outliers stem from a misalignment between template-based bias probes and LLM pretraining data due to markedness. Regardless of the underlying cause, these findings highlight the need to consider the impact that the use of bias probes relying on marked text has on the measurement of bias. In this case, such probes produce largely misleading results.

Assuming that linguistic markedness contributes significantly to the measurements in this work, which requires further investigation to confirm, several avenues for mitigating such effects when using template-based probes are worth exploring. LLMs trained on a more global representation of text, where majority

demographics differ and marked groups vary, could improve the robustness of such models when encountering explicit demographic descriptors. Another approach is to strive to control for the impact of markedness by designing evaluation setups that test both unmarked and explicitly marked versions of the same text (for example, comparing "a CEO" and "a White CEO") or by using neutral placeholders like [RACE] to isolate the impact of demographic terms and potentially correct for the flaws identified in this work. There are, however, challenges with this approach. One needs to identify which groups are considered unmarked from the "perspective" of the LLM, requiring detailed knowledge of the underlying pretraining data. In addition, the unassociated text is unlikely to solely represent the unmarked group, but rather a mix of representations. Ideally, artificial injection of demographic information would not be required. For example, the studies of Seyyed-Kalantari et al. (2020) and Sap et al. (2019) establish group membership through meta-data, self-identification, or classification techniques rather than explicitly in text. These methods avoid the out-of-domain nature of template-based examples of the kind studied here and do not see the unnatural patterns we observed. Future work will design experiments to validate the misalignment due to markedness conjecture and construct straightforward ways to mitigate such issues in LLMs.

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

# A  Fine-Tuning Hyperparameters

For completeness, we provide the full details of the hyperparameter tuning process used in the fine-tuning experiments. During fine-tuning, early stopping is applied based on validation loss. If no improvement in the loss is observed over a fixed number of steps, then training is stopped. An AdamW optimizer is used with default parameters, except for learning rate (LR) and weight decay (Loshchilov & Hutter, 2019). A hyperparameter study was performed to select the best early stopping threshold and LR for all models. For fully fine-tuned models, weight decay was also optimized.

The early stopping threshold was varied between five and seven steps. The learning rate (LR) was chosen from {1e-3, 3e-4, 1e-4, 3e-5, 1e-5}, and weight decay, when tuned, was selected from {1e-3, 1e-4, 1e-5, 1e-6}. For larger models, LoRA fine-tuning was applied with the rank parameter 8 on every non-embedding layer.

For RoBERTa 125M and 355M and OPT 125M and 350M, 15 training runs were performed, and the five models with the highest accuracy on the SST5 test set were retained. For the larger models, due to resource constraints, five models in total were trained for each model type. Table 1 summarizes the optimal hyperparameters selected for each model during fine-tuning.

Table 1: Hyperparameters used for model fine-tuning.

| Model | Early stop threshold | LR | Weight decay |
|---|---|---|---|
| RoBERTa-125M | 7 | 1e−5 | 1e−5 |
| RoBERTa-355M | 7 | 1e−5 | 1e−5 |
| OPT-125M | 7 | 1e−5 | 1e−5 |
| OPT-350M | 7 | 1e−5 | 1e−3 |
| OPT-1.3B | 5 | 1e−4 | 1e−4 |
| OPT-6.7B | 5 | 1e−4 | 1e−4 |
| OPT-13B | 5 | 1e−4 | 1e−4 |
| Llama-2-7B | 5 | 1e−4 | 1e−4 |
| Llama-2-13B | 5 | 1e−4 | 1e−4 |
| Llama-3-8B | 5 | 1e−4 | 1e−3 |
| Mistral-7B | 5 | 3e−5 | 1e−3 |

# B  Prompt Templates and Other Details

This section includes the templates used in the prompting approach. Each subsection corresponds to a different template. For CoT prompting, inference batches are limited to size 4 due to higher computational demands, whereas batch sizes of 16 are used in other settings.

## B.1  Zero-Shot Prompt Template

The zero-shot prompt template is displayed below with additional formatting for readability. The component in angled brackets is where each sample to be classified is inserted. The models begin generation at [*LM Generation*].

**Text:** ⟨Text to classify⟩
**Question: Is the sentiment of the text negative, neutral, or positive?**
**Answer: The sentiment is** [*LM Generation*]

### B.2 Few-Shot Prompt

Below is the few-shot template. For the few-shot prompt templates, nine labeled examples are prepended to the prompt, following the template style. The models begin generation at [*LM Generation*].

**Text: Example 1 from either SST5 or SemEval**
**Question: What is the sentiment of the text?**
**Answer: Negative.**

**...**

**Text: Example 9 from either SST5 or SemEval**
**Question: What is the sentiment of the text?**
**Answer: Positive.**

**Text:** ⟨Text to classify⟩
**Question: What is the sentiment of the text?**
**Answer:** [*LM Generation*]

### B.3 Zero-Shot CoT Prompt

Zero-shot CoT uses two prompt templates in sequence. In the first step, the model is provided the text to classify and asked about the corresponding sentiment. The traditional "trigger" sentence "Let's think step by step" is used to encourage the model to generate reasoning prior to answering the question. The template appears below.

**Text:** ⟨Text to classify⟩
**Question: Is the sentiment of the text negative, neutral, or positive?**
**Reasoning: Let's think step by step.** [*LM Generation*]

In the second step of zero-shot CoT, the reasoning generation is appended to the first prompt along with the answer completion text displayed in the template below. At this stage, the model is expected to generate an answer to be extracted.

**Text:** ⟨Text to classify⟩
**Question: Is the sentiment of the text negative, neutral, or positive?**
**Reasoning: Let's think step by step.** ⟨Generation from previous step⟩
**Answer: Therefore, from negative, neutral, or positive, the sentiment is** [*LM Generation*]

Table 2: Accuracy statistics on the Amazon dataset for fine-tuning experiments across model types and sizes. Bold numbers indicate the best accuracy achieved within each model family.

| Model | Size | Mean Accuracy | Standard Deviation |
|---|---|---|---|
| RoBERTa | 125M | **0.635** | 0.036 |
| | 350M | 0.624 | 0.027 |
| OPT | 125M | 0.687 | 0.080 |
| | 350M | 0.692 | 0.039 |
| | 1.3B | **0.739** | 0.020 |
| | 6.7B | 0.737 | 0.014 |
| Llama-2 | 7B | 0.513 | 0.089 |
| | 13B | **0.647** | 0.006 |
| Llama-3 | 8B | **0.822** | 0.035 |
| Mistral | 7B | **0.740** | 0.005 |

## C Fine-tuning and Prompting Accuracy

Tables 2 and 3 present the average classification accuracy and standard deviation for the fine-tuning and prompting approaches on the Amazon dataset, respectively. Generally, prompt-based classification accuracy is lower than that of fine-tuning. There is also notable variability in model classification accuracy on the template-based probe dataset as a whole, with newer models tending to produce better performance. However, the trends associated with the measured FPR gaps, especially for the White group, are largely consistent, regardless of these variations. Do to this consistency, the differences in model accuracy are unlikely to be a primary contributor to the observed behavior.

Table 3: Model accuracy and standard deviation (in parentheses) on the Amazon dataset for prompting experiments across model types. Bold numbers indicate the best accuracy achieved for each model.

| Prompt Type | Zero-shot | Zero-shot CoT | SemEval 9-shot | SST5 9-shot |
|---|---|---|---|---|
| OPT-6.7B | 0.451 (0.002) | – | **0.482** (0.009) | 0.433 (0.024) |
| Llama-2-7B | 0.483 (0.002) | 0.492 (0.003) | **0.654** (0.037) | 0.616 (0.028) |
| Llama-3-8B | 0.600 (0.003) | 0.539 (0.001) | 0.683 (0.017) | **0.716** (0.024) |
| Mistral-7B | 0.502 (0.003) | 0.517 (0.003) | **0.700** (0.045) | 0.682 (0.025) |
| Gemma-7B | 0.830 (0.001) | 0.777 (0.003) | **0.854** (0.020) | 0.804 (0.031) |
| Qwen-2.5-7B | 0.899 (0.001) | 0.823 (0.002) | **0.923** (0.016) | 0.906 (0.001) |

Table 4: Models ranked by average gap spans across datasets for Negative- and Positive-Sentiment FPR when fine-tuning. For a given type of FPR, gap spans are computed as the largest difference between any two mean FPR gaps across groups. The larger this span, the greater the difference in Negative- or Positive-Sentiment FPR between groups and the less invariant the model is to group substitution.

| Rank | Model | Mean Negative-Sentiment FPR Gap Span | Model | Mean Positive-Sentiment FPR Gap Span |
|---|---|---|---|---|
| 1 | Llama-2-13B | 0.207 | RoBERTa-355M | 0.154 |
| 2 | RoBERTa-355M | 0.198 | RoBERTa-125M | 0.152 |
| 3 | Llama-2-7B | 0.144 | OPT-13B | 0.144 |
| 4 | RoBERTa-125M | 0.126 | Llama-2-13B | 0.143 |
| 5 | OPT-350M | 0.118 | Mistral-7B | 0.141 |
| 6 | OPT-1.3B | 0.104 | OPT-125M | 0.136 |
| 7 | OPT-6.7B | 0.081 | Llama-2-7B | 0.133 |
| 8 | OPT-13B | 0.056 | OPT-350M | 0.132 |
| 9 | OPT-125M | 0.039 | OPT-1.3B | 0.128 |
| 10 | Mistral-7B | 0.032 | OPT-6.7B | **0.089** |
| 11 | Llama-3-8B | **0.020** | Llama-3-8B | **0.089** |

## D Gap Differences Across Models

For each of the fine-tuned models, across the different datasets, an FPR-gap span is calculated. For a given type of FPR, Negative- or Positive-Sentiment gap spans are computed as the largest difference between any two mean FPR gaps for the groups. This quantifies how large the particular FPR disparities for a given model and dataset are between groups. The larger this span, the greater the difference in Negative- or Positive-Sentiment FPR between groups and the less invariant the model is to overall group substitution. Table 4 displays the FPR gap spans for each model, averaged over the three datasets. In computing the spans, the gap for the White group is part of the span extremes 58% of the time for Negative-Sentiment FPR and 100% of the time for Positive-Sentiment FPR. That is, the gap computed for the White group often constitutes one of the largest gap magnitudes.

From the table, it is clear that the RoBERTa and Llama-2 models have consistently large spans for both types of FPR gap. On the other hand, Llama-3-8B, the most recent model studied with fine-tuning, has the smallest average gap spans in both categories. Another recent model, Mistral-7B, demonstrates a small average Negative-Sentiment FPR gap span, suggesting that more recent LLMs may be slightly less affected by issues with the template-based probes. It is interesting to note that the distribution of spans for Positive-Sentiment FPR gaps are more uniformly distributed between models than the Negative-Sentiment counterpart.

Table 5: Statistics for the three template-based probing datasets, Amazon, Regard, and NS-Prompts. The label counts and distributions are reported for each racial group across datasets. The label distributions are constant between groups, and thus reported below the label totals. Note that all examples from the NS-Prompts dataset have a neutral label by construction.

| Dataset | Race | Label Counts | | | Total | Fraction |
|---|---|---|---|---|---|---|
| | | Negative | Neutral | Positive | | |
| Amazon | African American | 200 | 200 | 200 | 600 | 0.182 |
| | American Indian | 300 | 300 | 300 | 900 | 0.273 |
| | Asian | 100 | 100 | 100 | 300 | 0.091 |
| | Hispanic | 200 | 200 | 200 | 600 | 0.182 |
| | Pacific Islander | 200 | 200 | 200 | 600 | 0.182 |
| | White | 100 | 100 | 100 | 300 | 0.091 |
| | Total | 1100 | 1100 | 1100 | 3300 | |
| | Distribution | 0.333 | 0.333 | 0.333 | | |
| Regard | African American | 440 | 220 | 350 | 1010 | 0.182 |
| | American Indian | 660 | 330 | 525 | 1515 | 0.273 |
| | Asian | 220 | 110 | 175 | 505 | 0.091 |
| | Hispanic | 440 | 220 | 350 | 1010 | 0.182 |
| | Pacific Islander | 440 | 220 | 440 | 1010 | 0.182 |
| | White | 220 | 110 | 175 | 505 | 0.091 |
| | Total | 2420 | 1210 | 1925 | 5555 | |
| | Distribution | 0.437 | 0.218 | 0.347 | | |
| NS-Prompts | African American | 0 | 8880 | 0 | 8880 | 0.182 |
| | American Indian | 0 | 13320 | 0 | 13320 | 0.273 |
| | Asian | 0 | 4440 | 0 | 4440 | 0.091 |
| | Hispanic | 0 | 8880 | 0 | 8880 | 0.182 |
| | Pacific Islander | 0 | 8880 | 0 | 8880 | 0.182 |
| | White | 0 | 4440 | 0 | 4440 | 0.091 |
| | Total | 0 | 48840 | 0 | 48840 | |
| | Distribution | 0.0 | 1.0 | 0.0 | | |

## E  Template-Based Dataset Statistics

In this appendix, some statistics and distribution information about the template-based probing datasets described in Section 3.1 are presented. Table 5 summarizes the label counts across datasets and broken down by racial groups. Note that the label distributions are constant between groups. For example, the label distribution of $(0.333, 0.333, 0.333)$ holds for each protected group as well as the aggregated label distribution. Further, it is important to recall that the models are not trained on these datasets and only perform classification inference to produce the results of Section 4. For the Amazon dataset, the labels are balanced over the entire dataset and within each individual group. The distribution for the Regard dataset is also fairly balanced with somewhat fewer neutral examples. Finally, as discussed in Section 3.1.2, all examples for the NS-Prompts dataset have a neutral label.

## F  Supporting Results for Unmarked Groups in Other Sensitive Attributes

This work primarily focuses on unexpected measurements with respect to the sensitive attribute of race. However, the conjecture that markedness causes, or at least contributes to, the observed irregularities when using template-based probes would, in theory, generalize beyond racial demographics. For example, as discussed in Section 5.1, several previous studies have noted that gender-based markedness is also prevalent in web data and influences model behaviors. In such settings, male gender represents the unmarked default. Furthermore, linguistic markedness extends to sexuality, with heterosexuality comprising the predominant unmarked demographic (Zerubavel, 2018).

With this in mind, two additional experiments are conducted using the prompting approaches described in Section 3.4 and the Amazon dataset (Czarnowska et al., 2021). The Amazon dataset includes templates for sensitive attributes beyond race, including sexuality and gender, with the same structure described in Section 3.1.1. Prompt-based classification is applied to these templates, and the same FPR gaps are measured across various protected groups within the sensitive attributes.

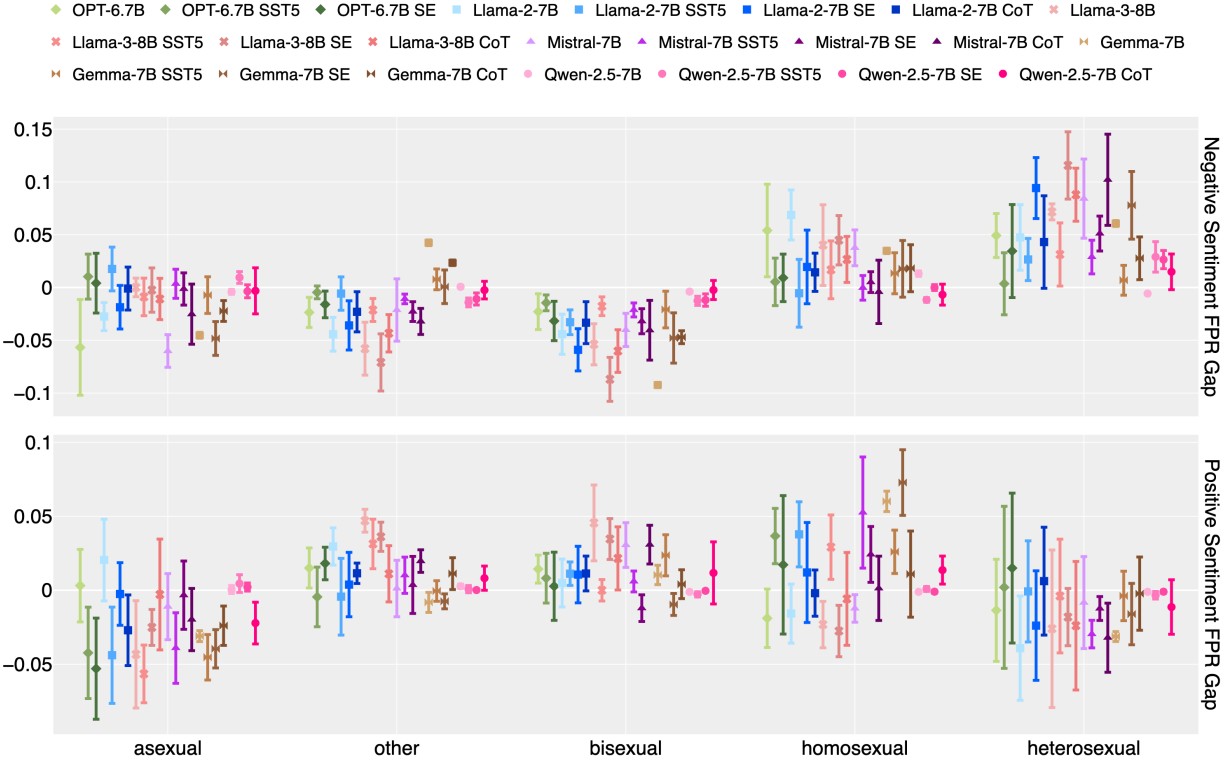

Figure 5: Negative- and Positive-Sentiment FPR gaps for protected group variations within the sensitive attribute of Sexuality as measured by the Amazon dataset. In the legend, model names without a suffix indicate zero-shot prompting. SST5 and SE indicate 9-shot prompts with examples drawn from the SST5 and SemEval datasets, respectively.

The results are shown in Figures 5 and 6. For sexuality, the traditionally unmarked group, heterosexual, reflects a similar FPR-gap pattern to the White group in the main results. That is, positive, and often statistically significant, Negative-Sentiment FPR gaps and negative Positive-Sentiment FPR gaps are found. Moreover, these patterns correlate to traditionally underprivileged groups in homosexual and asexual orientation for the respective FPR-gap types. When considering gender, the templates associated with male gender also produce positive and negative gaps for Negative- and Positive-Sentiment FPR, respectively, though the frequency of statistical significance is slightly reduced.

As noted in Section 5.1, additional experimentation is required to confirm that markedness is a driving factor for the results presented in this paper. Nonetheless, the empirical results in this section reinforce the observation that template-based probes produce imperfect measures of bias in LLMs and that these imperfections appear to affect measurements associated with traditionally unmarked groups, even beyond of the sensitive attribute of race.

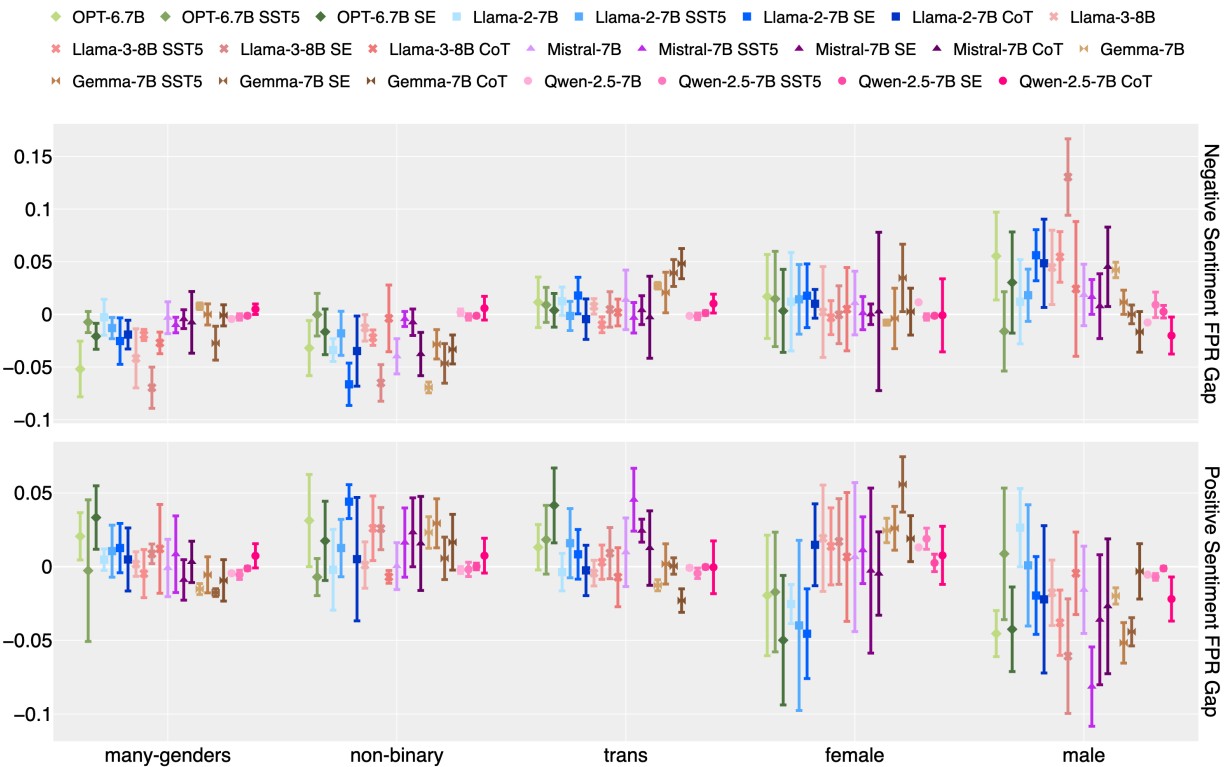

Figure 6: Negative- and Positive-Sentiment FPR gaps for protected group variations within the sensitive attribute of Gender as measured by the Amazon dataset. In the legend, model names without a suffix indicate zero-shot prompting. SST5 and SE indicate 9-shot prompts with examples drawn from the SST5 and SemEval datasets, respectively.

