# OpenReview forum: "Template-Based Probes Are Imperfect Lenses for Counterfactual Bias Evaluation in LLMs"
_TMLR — Accepted by TMLR_

### Review · Reviewer_5Rm5 · 2025-09-24

**Summary Of Contributions:**

Summary:

This paper critically examines template-based counterfactual bias evaluation in LLMs. The authors conduct experiments on three datasets (Amazon, NS-Prompts, and Regard), used as benchmarking templates, and assess a diverse set of model families and sizes, including RoBERTa, OPT, Llama-2/3, and Mistral. Across settings, the authors consistently observe that text templates containing the term \textit{white} are disproportionately classified as negative. The authors argument/hypothesize that this pattern does not represent the model bias, but rather arises as an artifact of linguistic markedness, given that \textit{white} typically functions as the unmarked demographic attribute in natural text. The study thereby underscores important limitations of template-based probes serving as tools for bias evaluation.

Strengths:
* Importance of the topic: Bias evaluation is a critical and highly relevant issue.
* Importance of the experimental work: Assessing the reliability of bias evaluation methods is a fundamental aspect for bias evaluation in ML.
* Experimental scope: The experiments span multiple model families and sizes, and benchmarking values are derived by using multiple complementary approaches, including both fine-tuning to a downstream sentiment analysis task and prompting.
* Interesting aspects/assumptions: Connecting possible template-based bias evaluation flaws to the linguistic aspect of markedness.

Weaknesses:
* The explanation for the findings (markedness) is presented by the authors as an hypthesis and has not been empirically tested.
* The datasets used for benchmarking could be described in greater detail, particularly with respect to their size (size of the benchmarking set per label) and label distribution.

**Audience:**

Yes

**Audience Explanation:**

Bias evaluation is central to the ML community, and template-based probes are a common approach for testing models after training. Demonstrating potential weaknesses of such methods provides valuable insights for the audience.

**Claims And Evidence:**

Yes

**Claims Explanation:**

The empirical findings (FPR gaps for templates containing the term 'white') are well supported across multiple settings (models, benchmarking data and evaluation scenarios). Nonetheless, the causal explanation regarding markedness is presented as a hypothesis rather than a validated mechanism, which is reasonable, since the authors explicitly acknowledge this limitation and clearly highlight this claim as an assumption.

**Requested Changes:**

Kindly address the weaknesses listed above.
1) Statistical description of benchmarking datasets:
* It would be helpful if the paper included a brief statistical description of the probing datasets (e.g., total number of samples, distribution across sentiment labels). This would make it easier for readers to interpret the reported FPR gaps and understand how dataset composition might influence the results.

2) Evaluation or discussion of markedness in training data
* If feasible, a simple analysis (e.g., string matching frequency) in pre-training and/or fine-tuning corpora could empirically support the findings.
* If that is not feasible due to e.g. data availability, it may still be useful to explicitly acknowledge the reasons and clarify why markedness was not addressed or is considered out of scope, while emphasizing that it requires further empirical validation in future work.

Minor (optional) suggestions for stylistic/editorial improvements:
* [1 Introduction] Syntax: 'a wide range' is usually treated as singular ('is considered') rather than plural ('are considered').
* [3.4 Prompting Experimental Setup] Figure positioning: Figure 1 seems to be mispositioned. It is appearing in the before going section of '3.4 Prompting Experimental Setup' while it is belonging to '4.1 Fine-Tuning Results'.
* [3.1.3 Regard Dataset] Formatting: The way labels are displayed in the line 'Labels: Regard: Negative. Sentiment: Positive.' could be streamlined. Rephrasing or reformatting would avoid the repeated colon.
* [4.1 Fine-Tuning Results] Formatting: The figures are very informative. It might make them even more effective if they were displayed at a larger size, for example using the full text width, so that they are easier to read.

---

> ### Author Response · Authors · 2025-10-30
>
> Thank you for your efforts with our manuscript and thoughtful review. We appreciate your comments on the importance and interesting nature of our experiments. Below is an itemized list of your requested changes. Each is followed by a discussion and description of how they were addressed in revising the paper. These changes have helped us improve the paper and the presentation of our results. Thank you again.
>
> **It would be helpful if the paper included a brief statistical description of the probing datasets (e.g., total number of samples, distribution across sentiment labels). This would make it easier for readers to interpret the reported FPR gaps and understand how dataset composition might influence the results.**
>
> Thank you for this suggestion. We agree that this would be helpful to an interested reader. We have revised the paper to include an Appendix entitled “Template-Based Dataset Statistics” which summarizes key statistics about each of the datasets in this work (Amazon, NS-Prompts, and Regard). These statistics include label counts and distributions across datasets and broken down by racial groups. The main table is approximately reproduced below for convenience.
>
> | Dataset        | Race             | Negative | Neutral | Positive | Total | Fraction |
> | -------------- | ---------------- | -------- | ------- | -------- | ----- | -------- |
> | Amazon  | African American | 200      | 200     | 200      | 600   | 0.182    |
> |                | American Indian  | 300      | 300     | 300      | 900   | 0.273    |
> |                | Asian            | 100      | 100     | 100      | 300   | 0.091    |
> |                | Hispanic         | 200      | 200     | 200      | 600   | 0.182    |
> |                | Pacific Islander | 200      | 200     | 200      | 600   | 0.182    |
> |                | White            | 100      | 100     | 100      | 300   | 0.091    |
> |                | Total        | 1100     | 1100    | 1100     | 3300  |          |
> |                | Distribution | 0.333    | 0.333   | 0.333    |       |          |
> | Regard   | African American | 440      | 220     | 350      | 1010  | 0.182    |
> |                | American Indian  | 660      | 330     | 525      | 1515  | 0.273    |
> |                | Asian            | 220      | 110     | 175      | 505   | 0.091    |
> |                | Hispanic         | 440      | 220     | 350      | 1010  | 0.182    |
> |                | Pacific Islander | 440      | 220     | 440      | 1010  | 0.182    |
> |                | White            | 220      | 110     | 175      | 505   | 0.091    |
> |                | Total        | 2420     | 1210    | 1925     | 5555  |          |
> |                | Distribution | 0.437    | 0.218   | 0.347    |       |          |
> | NS-Prompts | African American | 0        | 8880    | 0        | 8880  | 0.182    |
> |                | American Indian  | 0        | 13320   | 0        | 13320 | 0.273    |
> |                | Asian            | 0        | 4440    | 0        | 4440  | 0.091    |
> |                | Hispanic         | 0        | 8880    | 0        | 8880  | 0.182    |
> |                | Pacific Islander | 0        | 8880    | 0        | 8880  | 0.182    |
> |                | White            | 0        | 4440    | 0        | 4440  | 0.091    |
> |                | Total        | 0        | 48840   | 0        | 48840 |          |
> |                | Distribution | 0.0      | 1.0     | 0.0      |       |          |
>
> **If feasible, a simple analysis (e.g., string matching frequency) in pre-training and/or fine-tuning corpora could empirically support the findings.
> If that is not feasible due to e.g. data availability, it may still be useful to explicitly acknowledge the reasons and clarify why markedness was not addressed or is considered out of scope, while emphasizing that it requires further empirical validation in future work.**
>
> Unfortunately, the pre-training datasets for the models studied here are not publicly available for direct inspection. However, some existing work has established the prevalence of gender and racial markedness in English web data [1,2], on which LLMs are almost certainly pre-trained. Others have confirmed that this markedness is reflected in models trained on such data [2-4]. This prior work inspired our theory as to the cause of the abnormalities seen in the experiments.
>
> Ideally, to prove the conjecture of markedness as the mechanism introducing the distortions observed in this paper, we would pre-train an LLM on a “consistently marked” dataset and re-measure the presented gaps. Due to data availability and computational constraints, this valuable study is deferred to future work. As pointed out, we should have been more explicit about this deferral and its importance for future work to validate our hypothesis. We have now added an explicit discussion of this near the end of Section 5.1.

---

> > ### Author Response · Authors · 2025-10-30
> >
> > **[1 Introduction] Syntax: 'a wide range' is usually treated as singular ('is considered') rather than plural ('are considered').**
> >
> > **[3.4 Prompting Experimental Setup] Figure positioning: Figure 1 seems to be mispositioned. It is appearing in the before going section of '3.4 Prompting Experimental Setup' while it is belonging to '4.1 Fine-Tuning Results'.**
> >
> > **[3.1.3 Regard Dataset] Formatting: The way labels are displayed in the line 'Labels: Regard: Negative. Sentiment: Positive.' could be streamlined. Rephrasing or reformatting would avoid the repeated colon.**
> >
> > **[4.1 Fine-Tuning Results] Formatting: The figures are very informative. It might make them even more effective if they were displayed at a larger size, for example using the full text width, so that they are easier to read.**
> >
> > Thank you for pointing these out. We have incorporated each of your suggested edits above into the revised manuscript. Namely:
> >
> > 1. We have edited the syntax to match your suggestion.
> > 2. We have edited/repositioned Figure 1 to place it within Section “4.1 Fine-Tuning Results.” However, due to the size of the figures, precise placement is challenging. As such, we have allowed LaTeX to position Figures 3 and 4 slightly outside of their corresponding sections.
> > 3. The presentation of the Regard dataset labels has been reformatted to avoid the less refined structure as “Regard Label: Negative. Sentiment Label: Positive.”
> > 4. The figure sizes have been increased to full-text width to improve readability.
> >
> > [1] April H. Bailey, Adina Williams, and Andrei Cimpian. Based on billions of words on the internet, people=men. Science Advances, 8(13):eabm2463, 2022. doi: 10.1126/sciadv.abm2463. URL https://www.science.org/doi/abs/10.1126/sciadv.abm2463.
> >
> > [2] Robert Wolfe and Aylin Caliskan. Markedness in visual semantic ai. In Proceedings of the 2022 ACM Conference on Fairness, Accountability, and Transparency, FAccT ’22, pp. 1269–1279. Association for Computing Machinery, New York, NY, USA, 2022a. ISBN 9781450393522. doi: 10.1145/3531146.3533183. URL https://doi.org/10.1145/3531146.3533183.
> >
> > [3] Robert Wolfe and Aylin Caliskan. American == white in multimodal language-and-image ai. In Proceedings of the 2022 AAAI/ACM Conference on AI, Ethics, and Society, AIES ’22, pp. 800–812. Association for Computing Machinery, New York, NY, USA, 2022b. ISBN 9781450392471. doi: 10.1145/3514094.3534136. URL https://doi.org/10.1145/3514094.3534136.
> >
> > [4] Myra Cheng, Esin Durmus, and Dan Jurafsky. Marked personas: Using natural language prompts to measure stereotypes in language models. In Anna Rogers, Jordan Boyd-Graber, and Naoaki Okazaki (eds.), Proceedings of the 61st Annual Meeting of the Association for Computational Linguistics (Volume 1: Long Papers), pp. 1504–1532. Association for Computational Linguistics, Toronto, Canada, July 2023. doi: 10.18653/v1/2023.acl-long.84. URL https://aclanthology.org/2023.acl-long.84.

---

### Review · Reviewer_pWzh · 2025-09-30

**Summary Of Contributions:**

## Summary:

The paper discusses one of the limitations of some of the existing methods for bias assessment, namely template-based metrics. Template-based metrics quantify bias based on the change some measurable quantities (*e.g.* toxicity, sentiment, etc.) with inputs referencing varying races, genders, religions, etc. For example, models that do sentence completions with a higher toxicity for one race, compared to other races, are deemed biased against this race.

The argument made by the authors is: Due to the absence of markers when the referenced person is White (*e.g.,* it is more common to say “We hired a new CEO named Andy” than to say “We hired a new White CEO named Andy”), using template-based metrics yields incorrect results when referencing White people.

## Contributions:

1. Highlighting that template-based bias metrics provide inaccurate measures.
2. Conducting experiments on different models and datasets to support the their claim.

## Strengths:

1. The works is useful for the community as template-based metrics are widely used.
2. The writing and presentation of ideas in the paper are good.
3. The main message in the paper is timely and important for the community as template-based metrics are widely adopted.

## Weaknesses:
1. The experiments can be expanded to further solidify the authors' claims (see the Requested Changes section for more details).

**Audience:**

Yes

**Audience Explanation:**

Yes. Many individuals working on fairness would be interested in knowing the findings of this paper, as it questions a widely used set of metrics for bias measurement.

**Broader Impact Concerns:**

No concerns on ethical implications.

**Claims And Evidence:**

Yes

**Claims Explanation:**

Yes. Results on 3 datasets have emphasized the main argument by the authors by demonstrating that template-based metrics falsely show higher levels of bias against White people.

**Requested Changes:**

## Requested Changes:

1. It is important to include the authors’ thoughts on possible fixes for template-based metrics.

2. The study is only focused on race bias and White people specifically, although the same argument is applicable to other biases.
For example, it is more common to say “We need to hire a nurse” than to say “We need to hire a female nurse”.

3. While I generally agree with the paper’s main claims regarding the limitations in the current template-based metrics, I believe that more testing is needed. For example, can we show that by pre-training a model from scratch where all races are explicitly stated along the with the names of referenced individuals, such a limitation in template-based metrics would disappear?

4. Explaining how the regard being measured.

---

> ### Author Response · Authors · 2025-10-30
>
> We would like to thank you for your time and thorough review of our work. We are grateful for your comments on the strengths of our contributions. In addition, your feedback thus far has motivated a number of improvements to our work. Below is an itemized list of your requested changes, followed by a discussion and description of how they were addressed in revising the paper. We look forward to further discussing our work with you and any additional questions you may have.
>
> **It is important to include the authors’ thoughts on possible fixes for template-based metrics.**
>
> We agree that it would be helpful to discuss potential ways to account for the effects highlighted in this work.
>
> Assuming that the conjectured mechanism of markedness is the driving factor of the observed measurements, which requires further investigation to confirm, several avenues for mitigating such effects when using template-based probes exist. LLMs trained on a more global representation of text, where majority demographics differ and marked groups vary, could improve the robustness of such models when encountering explicit demographic descriptors. Another approach is to strive to control for sensitivity to unnatural phrasing and linguistic markedness by designing evaluation setups that test both unmarked and explicitly marked versions of the same text (for example, comparing “a doctor” and “a White doctor”) or by using neutral placeholders like [RACE] to isolate the impact of demographic terms and potentially correct for the flaws identified in this work. However, there are challenges with this approach. One needs to identify which groups are considered marked and unmarked from the “perspective” of the LLM, which requires some knowledge of the models underlying pretraining data. In addition, the unassociated text is unlikely to solely represent the unmarked default group, but rather a mix of representations. Ideally, benchmarks that establish group membership without explicit mention, through meta-data, would be used to circumvent the impact of markedness.
>
> We have included a discussion of the points above in Section 6 and look forward to working along these avenues in future work.
>
> **The study is only focused on race bias and White people specifically, although the same argument is applicable to other biases. For example, it is more common to say “We need to hire a nurse” than to say “We need to hire a female nurse”.**
>
> This is an important point, and we appreciate the suggestion. This study primarily considers bias measurements with respect to the sensitive attribute of race to ensure a deep and focused discussion. However, we believe that your observation provides important support for the conjecture that markedness plays a role in producing the gaps observed in the results. As such, we have conducted additional experiments considering the sensitive attributes of sexuality and gender, where heterosexuality and male gender have been shown to comprise the traditional unmarked groups [1, 2]. In performing prompt-based sentiment classification for sexuality and gender-based templates drawn from the Amazon dataset, strong FPR-gap patterns, similar to the White racial group in the main results, are observed for examples associated with heterosexual sexuality and male gender.
>
> These new results are now included in Appendix F of the paper and briefly discussed in Section 5.1. While more thorough experimentation is still required, we believe these results provide additional support for the conjecture that markedness causes, or at least contributes to, the observed irregularities when using template-based probes.
>
> **Explaining how the regard being measured.**
>
> In this work, regard inherits its definition from [3]. In their dataset, each sentence is given two labels: one for regard, which measures the sentiment with which the sentence talks about a demographic group (positive, neutral, or negative) in the first text phrase, and another for sentiment, which measures the overall emotional tone of the second phrase in the text. These labels were created through human annotation.
>
> We have added additional detail to Section 3.1.3 around how the Regard labels were derived to make the construction of this dataset, and how we used it in our experiments, more clear.

---

> > ### Author Response · Authors · 2025-10-30
> >
> > **While I generally agree with the paper’s main claims regarding the limitations in the current template-based metrics, I believe that more testing is needed. For example, can we show that by pre-training a model from scratch where all races are explicitly stated along the with the names of referenced individuals, such a limitation in template-based metrics would disappear?**
> >
> > We agree that testing the conjecture around the impact of markedness with a model pre-trained on data where racial identities are explicitly stated for all individuals would be an ideal next step. However, pre-training an LLM under these conditions requires access to a large pre-training corpus, substantial data curation efforts, and significant computational resources. As such, this valuable experiment is outside the scope of the current work. We are planning to undertake this effort in follow up work and look forward to sharing the results. A discussion recognizing the value of this experiment, its resource costs, and its place in our future work has been added near the end of Section 5.1
> >
> > [1] Linda R Waugh. Marked and unmarked: A choice between unequals in semiotic structure. Linguistics, 1982.
> >
> > [2] Eviatar Zerubavel. Taken for Granted: The Remarkable Power of the Unremarkable. Princeton University Press, 2018.
> >
> > [3] Emily Sheng, Kai-Wei Chang, Premkumar Natarajan, and Nanyun Peng. 2019. The Woman Worked as a Babysitter: On Biases in Language Generation. In Proceedings of the 2019 Conference on Empirical Methods in Natural Language Processing and the 9th International Joint Conference on Natural Language Processing (EMNLP-IJCNLP), pages 3407–3412, Hong Kong, China. Association for Computational Linguistics.

---

### Review · Reviewer_3c4z · 2025-10-24

**Summary Of Contributions:**

The manuscript examines social biases in language models' sentiment classification of templated sentences. Several experiments are done on different datasets across a range of language models, both generative and masked.

**Audience:**

No

**Audience Explanation:**

As explainer earlier, I am unsure about the novelty of the findings. The markedness is interesting, but no evidence points to this hypothesis and it is only presented in Section 5.

**Broader Impact Concerns:**

While the study itself focuses on bias measurements, I see no issue with the current setup.

**Claims And Evidence:**

No

**Claims Explanation:**

- The bias metric itself is not grounded in literature, while there is almost 20 years of research on classification with protected/sensitive attributes [1, 2]. Even for language models, there are plenty of metrics to choose from [4].
- The issues with templates have been well-studied by [3], albeit without the possible explanation of markedness and for BERT-style models. However, [3] tests this using different bias metrics and uses statistical tests and I see no reason their conclusions would not hold for generative LLMs.
-The "markedness" explanation is not actually investigated and feels added on after the fact. This feeling stems from the lack of explanation of markedness until the discussion and the framing of the results in the conclusion. I would at least move the explanation of this central term forward, as this is mainly a CS journal and linguistic or sociological terms might be less well known.
- There is a wide range of performance scores for different models. I am surprised by e.g. LLama 2 only scoring 0.55. Such a performance gap could be a cofounding factor for the presented results.
- I am unsure of the treatment of the "neutral" category from the sentiment classifications. It seems to be lumped together with "negative", except for the "negative sentiment FPR gap". This starts to get complicated to wrap my head around and a proper 3-class metric might have been more suitable.

[1] https://link.springer.com/content/pdf/10.1007/s10618-010-0190-x.pdf

[2] https://proceedings.neurips.cc/paper/2016/file/9d2682367c3935defcb1f9e247a97c0d-Paper.pdf

[3] https://aclanthology.org/anthology-files/pdf/naacl/2022.naacl-main.122.pdf

[4] https://aclanthology.org/anthology-files/pdf/emnlp/2024.emnlp-main.1207.pdf

**Requested Changes:**

**Required changes**
- All of the studies models are from 2023 or earlier, with the only exception llama 3. While I do appreciate the range of models, I think some modern LLMs should also be included. For these models, just prompting experiments are fine, since this is a more realistic setting for them anyways. In any case, models like OPT are honestly quite bad and I am not sure wether their results could generalize.
- The bias metric is very ad-hoc and not grounded in any literature. Since the entire paper hinges on this metric, some motivation would have been nice. Also, quite nitpicky, but the term "gap" for a metric that includes the $\bar{X}$ seems counter-intuitive.
- Rewrite the manuscript to focus properly on the "markedness", including some experiments that actually support this hypothesis.

**Clarifications**
- Please clarify if sentiment or regard is used for the regard dataset. Figure 3's caption and Section 3.1.3 seem contradictory.
- Section 3.4 discusses reasoning traces, but no reasoning LLMs are used (cfr. Deepseek R1). Is this referring to the chain-of-thought?

---

> ### Author Response · Authors · 2025-10-30
>
> Thank you for your thorough review of our manuscript and providing us the opportunity to improve the presentation of our results. We have made a number of revisions throughout the paper in response to your comments and suggestions. Below is an itemized list of your requested changes and clarifications, followed by a discussion and description of how they were addressed in revising the paper. Note that some related comments have been grouped together. Please do not hesitate to ask any additional questions you may have.
>
> **The bias metric itself is not grounded in literature, while there is almost 20 years of research on classification with protected/sensitive attributes [1, 2]. Even for language models, there are plenty of metrics to choose from [4].**
>
> **The bias metric is very ad-hoc and not grounded in any literature. Since the entire paper hinges on this metric, some motivation would have been nice. Also, quite nitpicky, but the term "gap" for a metric that includes the $\bar{X}$ seems counter-intuitive.**
>
> We appreciate the opportunity to better motivate our choice of bias metric. The M-gap metric defined in Section 3.2 is tightly related to the error rate equality difference metrics of [1] and, more specifically, False Positive Equality Difference (FPED). These metrics are also documented in the taxonomy of bias metrics in [2]. In our work, we make three small modifications to FPED to more closely examine the disparities across protected groups.
>
> 1. Rather than summing up contributions across groups, we examine differences individually.
> 2. We do not compute the absolute value of the difference to preserve the direction of the difference, as done in [4].
> 3. We compute the average false positive rates across groups, as the number of examples from each group is not the same.
>
> While we disagree that the bias metric used in this work is not grounded in previous literature, we did not do a good enough job demonstrating its relationship to previous work. In Section 3.2, where the metric is first defined, we now explicitly discuss the metrics relationship to FPED in [1,2]. This provides better motivation for the metric and properly grounds the choice in existing literature.
>
> With respect to the use of “gap,” the name was motivated by several existing metrics using the term [2, 4]. However, we are happy to rename the metric, if you feel strongly that it should be.
>
> **The issues with templates have been well-studied by [3], albeit without the possible explanation of markedness and for BERT-style models. However, [3] tests this using different bias metrics and uses statistical tests and I see no reason their conclusions would not hold for generative LLMs.**
>
> We agree that flaws in templates have been studied from different angles in previous work. The work of [3] does discuss flaws in template design, but does so from the perspective of intrinsic bias measurements. The primary conclusion of [3] is that intrinsic bias measurements do not correlate well with extrinsic (downstream task) bias measurements and that many intrinsic measures are not consistent with one another, potentially due to template design, among other factors.
>
> While the study does suggest that templates must be used carefully when applying intrinsic metrics, the focus of our work is quantifying how flaws in template-based probes affect extrinsic bias measurements. We show that template-based probes lead to systematic group-level difference errors. This is an effect that intrinsic metrics alone cannot capture. Our work also considers both masked language models (MLMs) and generative LLMs. While it may be intuitive that issues influencing MLMs might also affect generative LLMs, such models differ in many significant ways. As such, we believe there is notable merit in extending results to generative and modern LLMs, even if some results may exist for small MLMs.
>
> We recognize that [3] deserves a deeper discussion in this work. As such, we have added a more detailed overview of their results and conclusions to Section 2 (Related Work) and more clearly discuss its relationship to our work.

---

> > ### Author Response · Authors · 2025-10-30
> >
> > **The "markedness" explanation is not actually investigated and feels added on after the fact. This feeling stems from the lack of explanation of markedness until the discussion and the framing of the results in the conclusion. I would at least move the explanation of this central term forward, as this is mainly a CS journal and linguistic or sociological terms might be less well known.**
> >
> > Thank you for this suggestion. We have moved the definition of markedness to Section 2, as a special subsection of Related Work. This serves to concretely introduce the concept before pre-existing work related to markedness is discussed and prior to its reference in the results and discussion sections.
> >
> > In addition to this change, we have also included further experiments to support the conjecture that markedness plays an important role in the abnormal measurements seen in the results. In Appendix F, the prompting-based classification experiments are repeated for the sensitive attributes of sexuality and gender using templates drawn from the Amazon dataset. For these attributes, heterosexuality and male gender have been shown to comprise the traditional unmarked groups [5, 6]. Strong FPR-gap patterns, similar to the White racial group in the main results, are observed for examples associated with heterosexual sexuality and male gender. Further experiments are still required to fully confirm the markedness conjecture. However, we believe this work provides strong motivation for the hypothesis.
> >
> > **There is a wide range of performance scores for different models. I am surprised by e.g. LLama 2 only scoring 0.55. Such a performance gap could be a cofounding factor for the presented results.**
> >
> > While we are not certain of the metric to which 0.55 refers, we believe the performance gaps being referenced are those reported in Tables 2 and 3. Please correct us if this is not the intended set of results.
> >
> > There is a wide range of performance across models for both the fine-tuning and prompting-based results. While certain models exhibit lower accuracy on the template-based probe dataset as a whole, compared to others, the trends associated with the FPR-gaps for the White protected group are largely consistent. Because of this consistency, we do not believe the differences in model accuracy are a primary contributor to the observed behavior. However, we do think it is important to discuss this point more directly in the paper and have added a discussion of these points to Appendix C for better clarity.
> >
> > **I am unsure of the treatment of the "neutral" category from the sentiment classifications. It seems to be lumped together with "negative", except for the "negative sentiment FPR gap". This starts to get complicated to wrap my head around and a proper 3-class metric might have been more suitable.**
> >
> > In this work, we focus on disparities between protected groups in false positive rate (FPR). In the 3-way classification setting, these are incorrect predictions of one class, when the true label was one of the other two classes. For example, Negative-sentiment FPR is the rate at which samples labeled either positive or neutral are misclassified as negative. Similarly, Positive-sentiment FPR is the rate at which samples labeled either negative or neutral are misclassified as positive. These metrics are commonly used in multi-class settings.
> >
> > When considering Negative- or Positive-sentiment FPR, interpretation of errors is fairly straightforward. In the case of Negative-sentiment FPR, prediction errors deem a sample to be more negative than its true label (positive or neutral). On the other hand, Positive-sentiment FPR involves errors where the model casts a sample in a more positive light than its label (negative or neutral). Neutral-sentiment FPR is more convolved. Such errors correspond to neutral predictions for samples with either negative or positive labels. Errors are a mix of predictions viewing samples with negative labels more positively and predictions viewing samples with positive labels more negatively. This clouds interpretation of Neutral-sentiment FPR gaps without further decomposition of the metric. As such, results are limited to Negative- and Positive-sentiment FPR in this work.
> >
> > We have expanded our introduction of the bias metric in Section 3.2 to include components of the above discussion to more clearly motivate our choice to emphasize Negative- and Positive-sentiment FPR.

---

> > > ### Author Response · Authors · 2025-10-30
> > >
> > > **All of the studies models are from 2023 or earlier, with the only exception llama 3. While I do appreciate the range of models, I think some modern LLMs should also be included. For these models, just prompting experiments are fine, since this is a more realistic setting for them anyways. In any case, models like OPT are honestly quite bad and I am not sure wether their results could generalize.**
> > >
> > > We appreciate the suggestion. Two new models have been added to the prompting experiments, Gemma-7B Instruct and Qwen-2.5-7B Instruct. Classification accuracy with these models is markedly better than previous models. Nonetheless, these models display similar patterns for samples associated with White race to previous models. The size of the effect appears slightly smaller for Qwen-2.5, but remains present.
> > >
> > > Additionally, the newly included experiments with gender and sexuality templates, in Appendix F, have also been updated to include these models.
> > >
> > > Finally, we note that OPT does perform poorly, though still well above a random classifier, for the sentiment-classification task through prompting. However, through fine-tuning, it performs the task quite well. Regardless of performance, the model demonstrates FPR-gap patterns consistent with other models.
> > >
> > > **Rewrite the manuscript to focus properly on the "markedness", including some experiments that actually support this hypothesis.**
> > >
> > > We are happy to hear that you find the markedness hypothesis interesting. As noted in Section 6, experiments designed to fully confirm the role of markedness in the observed anomalies are being actively planned in future work. The primary contribution of this research is the large-scale quantification of consistent flaws in counterfactual bias evaluation in LLMs through template-based probes. These flaws have not been rigorously measured in previous work and are highly important, given the widespread use of templates for bias measurement.
> > >
> > > In this work, the markedness conjecture is offered as a potential factor contributing to the observed flaws and as a motivator of future work on this topic. While we present some experiments that peripherally support the conjecture, more work is needed to confirm the role of markedness.
> > >
> > > **Please clarify if sentiment or regard is used for the regard dataset. Figure 3's caption and Section 3.1.3 seem contradictory.**
> > >
> > > Thank you for pointing this out. In our original description, we were not as clear as we could have been in describing the construction of the Regard templates that are used. The original labels for the regard and sentiment phrases are both sentiment-based, but only the regard phrase includes explicit demographic information. As such, that phrase is retained, along with its sentiment label. So we refer to the label as “sentiment” in the remainder of the manuscript. To provide better clarity, we have included elements of the discussion above into Section 3.1.3.
> > >
> > > **Section 3.4 discusses reasoning traces, but no reasoning LLMs are used (cfr. Deepseek R1). Is this referring to the chain-of-thought?**
> > >
> > > Yes. In the original work introducing zero-shot CoT, the intermediate generations were referred to as “reasoning paths,” but, in our experience, “reasoning traces” has become the more commonly used term. If you have an alternative phrase that better fits this setting, please let us know.
> > >
> > > [1] Lucas Dixon, John Li, Jeffrey Sorensen, Nithum Thain, and Lucy Vasserman. Measuring and mitigating unintended bias in text classification. In Proceedings of the 2018 AAAI/ACM Conference on AI, Ethics, and Society, AIES ’18, pp. 67–73, New York, NY, USA, 2018. Association for Computing Machinery.
> > >
> > > [2] Paula Czarnowska, Yogarshi Vyas, and Kashif Shah. Quantifying social biases in NLP: A generalization and empirical comparison of extrinsic fairness metrics. Transactions of the Association for Computational Linguistics, 9:1249–1267, 2021.
> > >
> > > [3] Pieter Delobelle, Ewoenam Tokpo, Toon Calders, and Bettina Berendt. Measuring fairness with biased rulers: A comparative study on bias metrics for pre-trained language models. In Marine Carpuat, Marie-Catherine de Marneffe, and Ivan Vladimir Meza Ruiz (eds.), Proceedings of the 2022 Conference of the North American Chapter of the Association for Computational Linguistics: Human Language Technologies, pp. 1693–1706. Association for Computational Linguistics, Seattle, United States, July 2022.
> > >
> > > [4] Laleh Seyyed-Kalantari, Guanxiong Liu, Matthew McDermott, Irene Y Chen, and Marzyeh Ghassemi. Chexclusion: Fairness gaps in deep chest x-ray classifiers. In BIOCOMPUTING 2021: proceedings of the Pacific symposium, pp. 232–243. World Scientific Publishing Company, 2020.
> > >
> > > [5] Linda R Waugh. Marked and unmarked: A choice between unequals in semiotic structure. Linguistics, 1982.
> > >
> > > [6] Eviatar Zerubavel. Taken for Granted: The Remarkable Power of the Unremarkable. Princeton University Press, 2018.

---

### Decision · Action_Editor_nBfG · 2026-01-12

**Recommendation:** Accept as is

**Additional Comments:**

Overall, I think the authors did a thorough job of addressing reviewer comments in their rebuttal and revisions. All reviewers were in favor of acceptance after the rebuttal, whereas one was clearly against (in my view) before the rebuttal. Among changes not already mentioned above, relationships with prior work (particularly the PFED metric and template-based probing of intrinsic bias) are now better discussed, the definition of class-specific FPR has been clarified, and dataset statistics have been added and figures improved to help interpret results. I see no further issues and recommend acceptance as-is.

**Audience:**

Yes

**Audience Explanation:**

Bias evaluation of LLMs is a well-established topic of study and template-based probing is a widely used technique for bias evaluation. Thus, the paper should be of high interest to many researchers in this area.

**Claims And Evidence:**

Yes

**Claims Explanation:**

This submission addresses bias evaluation of LLMs using template-based probes, which prompt LLMs using a template that explicitly states membership in a social group (e.g. race). The main finding is that there are consistent gaps in false positive rates (FPRs) between texts mentioning White race versus other races, which together suggest that LLMs have a bias toward negative sentiment for White race. The reviewers found this conclusion to be well-supported across multiple LLM families, template-based probing datasets, and for both prompted and fine-tuned LLMs. During the rebuttal period, the authors further strengthened the evidence by adding Gemma and Qwen-2.5 models to address a comment about the tested LLMs being out of date.

The submission puts forth markedness (explicitly mentioning a social group or not) as a hypothesis to explain the findings, and makes clear that it is still a hypothesis that needs further testing. The authors made progress in this direction during the rebuttal period by repeating their experiment using sexuality and gender as sensitive attributes, with consistent results. The manuscript was also improved by adding discussion of the challenges of conducting further experiments to test the hypothesis, as well as by introducing markedness earlier in the paper.